# Mucosal Melanoma Clinical Management and Prognostic Implications: A Retrospective Cohort Study

**DOI:** 10.3390/cancers16010227

**Published:** 2024-01-03

**Authors:** Laia Clavero-Rovira, Álvaro Gómez-Tomás, Patricia Bassas-Freixas, Domingo Bodet, Berta Ferrer, Javier Hernández-Losa, Eva Muñoz-Couselo, Assumpció Pérez-Benavente, Vicente García-Patos, Carla Ferrándiz-Pulido

**Affiliations:** 1Department of Dermatology, University Hospital Vall d’Hebron, 08035 Barcelona, Spain; laia.clavero@vallhebron.cat (L.C.-R.); alvaro.gomez@vallhebron.cat (Á.G.-T.); patricia.bassas@vallhebron.cat (P.B.-F.); domingo.bodet@vallhebron.cat (D.B.); vicente.garciapatos@vallhebron.cat (V.G.-P.); 2Facultad de Medicina, Universitat Autònoma de Barcelona, 08193 Bellaterra, Spain; assump.perez@vallhebron.cat; 3Department of Pathology, University Hospital Vall d’Hebron, 08035 Barcelona, Spain; berta.ferrer@vallhebron.cat (B.F.); javier.hernandez@vallhebron.cat (J.H.-L.); 4Department of Oncology, University Hospital Vall d’Hebron, 08035 Barcelona, Spain; eva.munozcouselo@vallhebron.cat; 5Unit of Gynecologic Oncology, Department of Obstetrics and Gynecology, University Hospital Vall d’Hebron, 08035 Barcelona, Spain

**Keywords:** mucosal melanoma, somatic mutations, prognostic factors, Breslow depth, immunotherapy, survival outcomes

## Abstract

**Simple Summary:**

Mucosal melanoma (MM) is a rare melanoma subtype that affects mucosal surfaces of the head and neck, anorectal region, and vulvovaginal area. Due to its rarity, the management, monitoring, and treatment of MM lack standardization, often relying on protocols designed for cutaneous melanoma (CM). In this retrospective, registry-based cohort study, we analyzed epidemiological and histological data together with the treatments administered to gain insights into the disease’s behavior, treatment patterns, and potential predictors of survival. Our findings reveal that lower disease stage, thinner Breslow depth, and surgical resection are associated with improved overall survival, while age, sex, radiotherapy, and *BRAF* mutant status do not significantly affect survival. Standard systemic management typically includes immunotherapy (anti-PD-1 or anti-PD-1 and anti-CTLA-4). For cases with *BRAF* or *c-KIT* mutations, targeted therapies may be considered. The prognosis is unfavorable, with a survival rate of less than 50% at 2 years.

**Abstract:**

Mucosal melanoma (MM) is an uncommon melanoma subtype affecting mucosal surfaces of the head and neck, anorectal region, and vulvovaginal area. We aimed to present our experience at a tertiary-level hospital regarding MM diagnosis, management, monitoring of progression, mutations, and outcome predictors. We performed a registry-based cohort study including MM cases diagnosed from 2012 to 2022 and retrospectively characterized somatic mutations on *BRAF*, *NRAS* and *c-KIT.* We employed Kaplan–Meier curves, log-rank tests, and Cox regression analysis to explore prognostic factors and survival outcomes in a cohort of 35 patients, mainly women (63%) with a median age of 70 years. Predominantly, MM occurred in the vulvovaginal region (48.6%). At diagnosis, 28.6% had lymph node involvement, and 31.4% also had distant metastasis. Mutations in *BRAF* and *c-KIT* were identified in 3/35 (9%) and 2/6 patients (33%), respectively. Surgery was performed in 71.4% of patients, and most received systemic treatment (65.7%). Lower disease stage, thinner Breslow depth, and surgical resection were associated with improved overall survival. Notably, age, sex, radiotherapy, and *BRAF* mutant status did not affect survival. Standard management typically involves immunotherapy. Cases with *BRAF* or *c-KIT* mutations may be considered for targeted therapies. Unfortunately, MM prognosis remains unfavorable, with a less than 50% survival rate at 2 years.

## 1. Introduction

Melanomas are malignant neoplasms originating from melanocytes, which arise from neural crest cells and undergo migration through embryonic mesenchyme to reach their ultimate location. Most melanocytes are found in the epidermis and dermis, but they can also be found in diverse extracutaneous sites including the ocular region, mucosal tissues, and leptomeninges [1,2].

Primary MM include melanomas occurring in the head and neck mucosa, anorectal region, vulvovaginal area, and urinary tract, ranked in order of frequency. MM accounts for 1% of all melanomas, and its incidence is stable, in contrast to CM, which is experiencing an increasing incidence [3,4,5,6].

MM tends to manifest at advanced ages in comparison to CM, with a median age of diagnosis at 70 years. Moreover, it is more frequently identified in women, primarily due to the prevalence of vulvovaginal melanoma, which represents the most common subtype affecting women. In men, the head and neck region emerge as the principal site of MM [1,4,5,7].

Risk factors contributing to the onset of MM have yet to be definitively established [2,8]. Unlike CM, there is no association between ultraviolet radiation exposure and the development of MM [3]. Previous studies have explored viral exposures such as human papillomavirus [9], human herpes virus [10], and polyomavirus [11], as well as exposure to formaldehyde [12], as potential risk factors. However, these factors are not widely regarded as significant causes of MM. In the case of oral MM, cigarette smoking has been proposed as a risk factor, as studies have indicated a higher prevalence of oral pigmented lesions among smokers [13].

The underlying mechanisms driving the pathogenesis of MM remain unclear. In contrast to CM, which commonly exhibits oncogenic mutations in *BRAF*, such mutations are rarely observed in MM. However, activating mutations of *c-KIT* are more frequently detected in MM [2,3,5,14,15,16].

The clinical manifestation of MM often lacks specificity and varies depending on the site of origin [1]. The absence of early and distinct indicators, along with the hidden locations of the disease, in addition to frequent nodal and/or metastatic involvement at the time of diagnosis, are common factors contributing to an unfavorable prognosis and diminished survival rates.

There are few studies describing both the clinical characteristics and management of MM. In this study, our main objective is to describe the management of MM in a tertiary-level hospital regarding the diagnosis, treatment, and monitoring of MM, while also describing the frequency of common mutations and predictors of outcome.

## 2. Materials and Methods

### 2.1. Study Design and Data Source

We conducted a retrospective and descriptive registry-based cohort study at Vall d’Hebron University Hospital (Barcelona, Spain), reviewing the Pathology Department’s database to identify all cases of MM diagnosed between 1 January 2012, and 31 December 2022, resulting in a total of 35 cases. The study included cases of MM located in the head and neck region (nasal cavity, paranasal sinuses, and oral cavity), vulvovaginal region, anorectal region, and the rest of the digestive tract. Duplicate cases were excluded, where multiple biopsies were taken from the same patient, along with metastatic melanomas from primary CM. Following a systematized protocol, demographic and clinical data (age, sex, race, immunosuppressive state and smoking habit, tumor location, first symptom, description of the primary lesion, staging and treatment modalities), histopathological variables such as thickness (i.e., Breslow depth according to the College of American Pathologists guidelines depending on tumor location), and therapeutic and follow-up information were recorded from each patient through chart review. Survival time was defined as the time from the date of diagnosis to the date of death caused by the disease or, in case of no death or death from some other cause, the date of last follow-up/date of death.

Additionally, somatic mutations were retrospectively analyzed in the 35 samples by real-time polymerase chain reaction (PCR) using the *BRAF/NRAS* mutation test LSR (Roche Diagnostics). DNA extraction was performed from formalin-fixed–paraffin-embedded (FFPE) blocks using a COBAS DNA Sample preparation kit.

Given the complexity in establishing a unified staging system across diverse anatomical sites, we categorized staging into three groups: localized disease (Localized), nodal dissemination (Nodal), and distant metastatic disease (Distant).

### 2.2. Statistical Analysis

Descriptive and univariate statistics were computed as customary; the sample size was modest and nonparametric tests were used. Clinical and histological characteristics of the tumors were compared by anatomical location and stage using the Fisher’s exact test and the Mann–Whitney U test, as appropriate, unless stated otherwise. Post hoc tests with adjusted *p*-values were carried out if the omnibus test was statistically significant.

Kaplan–Meier curves were generated to compare survival between selected clinical or histological characteristics, and log-rank tests were used to ascertain differences between these groups. Cox proportional-hazards models (hazard ratio (HR) (95% CI)) were used to estimate the risk of overall mortality. Age, sex, and disease stage at diagnosis (localized, nodal, and distant) were included as potential confounders in these models.

All tests were two-tailed and *p*-value < 0.05 was considered significant. Statistical analyses were performed using R version 4.2.2 (R Core Team, Vienna, Austria, 2023).

## 3. Results

### 3.1. Characteristics of Patients and Tumors at Baseline

A retrospective analysis was conducted from 35 patients diagnosed with MM with a median follow-up of 21 months and total person-time follow-up of 936 months. Most patients were female with an observed sex ratio of 1.7:1 (62.9% females versus 37.1% males; *p* = 0.001; see Table 1). The median age at initial diagnosis was 70 years, ranging from 44 to 95 years, with no significant differences between anatomical sites. The vulvovaginal region was the most frequent (48.6% among females), encompassing melanomas located in both the vulva (*n* = 12) and the vagina (*n* = 5). The head and neck region was the second most common location (28.6%, *n* = 10), including cases in the nasal cavity and paranasal sinuses (*n* = 5) and the oral cavity and lip mucosa (*n* = 5). Gastrointestinal tract melanomas, including the anorectal region (*n* = 7) and esophagus (*n* = 1), were observed in eight patients (22.8%). Two of the patients were immunocompromised due to lung transplantation.

The first symptom observed among all patients was attributed to the progressive enlargement of the primary tumor, with the sole exception of an individual where the clinical presentation encompassed perceived growth of the primary tumor along with regional node enlargement. Clinical images of two patients are displayed in Figure 1. In vulvovaginal cases, bleeding was the most prevalent symptom, followed by the vulvar or vaginal tumor observation. For anorectal cases, rectorrhagia emerged as the predominant clinical presentation, while esophageal case primarily manifested as dysphagia. In head and neck melanomas, we could not identify a predominant initial symptom; the presentation depended on the exact site of involvement, with epistaxis or nasal obstruction in cases involving the nasal cavity, as well as the presence of ulceration in melanomas arising in the oral cavity.

The predominant cause of death among deceased patients was attributed to the progression of MM, with only two exceptions: one case resulted from cardiovascular disease, while another was due to respiratory insufficiency.

### 3.2. Breslow and Staging

At diagnosis, 14 cases (40%) presented with localized disease, whereas 10 cases (28.6%) also demonstrated lymph node involvement, without concurrent distant metastases. Additionally, 10 cases (28.6%) displayed both lymph node involvement and concurrent distant metastases, while a patient presented with distant metastases in the absence of lymph node involvement. Lymph node involvement was detected in 17 out of 20 clinically or using imaging studies. Positron emission tomography/computerized tomography (PET/CT) scan was the most common modality (52.9%), followed by magnetic resonance imaging (MRI) and CT (each 20%). In three cases, the lymph node metastases were microscopically detected by selective sentinel lymph node biopsy (SLNB) (15%).

There were no differences observed across various tumor sites with respect to age, disease stage, lymph node involvement, or distant metastases (Table 1), although a significant association was observed between age at diagnosis and disease stage, with younger age associated with more advanced stages (Appendix A). The median age for patients with localized disease was 83 years whilst for advanced and metastatic was 67 and 63 years, respectively (*p* = 0.001). We also observed a statistically significant relationship between disease stage and smoking habit, with most patients in the metastatic group being smokers (90% vs. 33% and 39% in the localized and nodal stages, respectively, *p* = 0.02, Appendix A).

The Breslow depth was determined in 19 out of 35 patients, with a median value of 5.5 mm. Median Breslow depth increased with disease stage, although these differences were not statistically significant (Appendix A).

### 3.3. Mutations

Among 35 MM cases, *BRAF* mutations were detected in three samples (8.6%, 95% CI: 1.8–23%), localized in exon 15 (p.V600D and p.V600E) and exon 11 (p.G466E). *NRAS* mutations were detected in two patients (5.7%, 95% CI: 0.7–19.2%), with changes at p.G13X and p.Q61X codons. The majority of cases (85.7%, 95% CI: 70–95%) were wild-type for both *BRAF* and *NRAS*. In six cases where *c-KIT* mutations were investigated, we found two positive cases (p.Y553C and p.Y578H) (33.3%, 95% CI: 4–77%) using a custom NGS panel. Additionally, a single *TP53* (p.A159V) mutation was detected. No differences were found between MM location and *BRAF* mutation rate (*p* = 0.782) or between staging and *BRAF* mutation (*p* = 1; Table 1).

### 3.4. Management

In our study, the disease management strategy was meticulously determined by expert committees, taking into account available guidelines, patient and tumor characteristics, and disease staging. This comprehensive approach often involved a combination of surgical interventions, radiotherapy, and tailored systemic therapies (see Table 2).

In the majority of patients, regardless of anatomical localization, tumor excision through surgery was the preferred approach (71.4%) (Table 2). Radiotherapy was not as common, employed in just 22.9% of patients. Systemic treatment was administered to 65.7% of patients, the majority (60.9%) receiving in an adjuvant setting after surgery, while nine patients received systemic treatment as their first option, either due to the extent of the disease or the impracticability of surgical intervention.

In 34.8% of patients, the most frequently used first-line systemic therapeutic agents across the three distinct anatomical localizations were the anti-programmed cell death-1 (anti-PD-1) immune checkpoint inhibitors (i.e., nivolumab or pembrolizumab). The predominant second-line choice was CTLA-4 immune checkpoint inhibitor (ipilimumab), utilized in 46.2% of patients, alongside with being the predominant third-line treatment option (33.3%). Table 2 provides an overview of the diverse treatments utilized, including their combinations.

### 3.5. Survival Outcomes

Investigating the survival outcomes in our cohort, we observed a 2-year survival rate of 43.5% (95% CI 29.4–64.2%) and a 5-year survival rate of 23.5% (95% CI 10.8–51.5%) (Figure 2). Among patients who died during the study period, the median time until this event occurred was 13 months.

Kaplan–Meier curves showed improved survival in patients with Breslow depths of less than 5 mm (*p* = 0.049, vs. ≥5 mm) (Figure 2). In multivariate Cox models, a greater Breslow depth was associated with a higher risk of death after adjusting for age, sex, and disease stage (HR: 1.51 95% CI: 1.13–2.01, *p* = 0.001) (Table 3). Different locations of MM (vulvovaginal, anorectal, or within the gastrointestinal tract) did not exhibit a definitive association with overall survival; however, the vulvovaginal subtype appeared to demonstrate a slightly more favorable prognosis. (Figure 2). When comparing vaginal and vulvar localization, our analysis of Kaplan–Meier survival curves revealed a significantly lower survival rate in cases of vaginal MM at 12 months, with a rate of 40%, in contrast to a notably higher survival rate of 92% observed in vulvar MM cases (*p*-value from the log-rank test = 0.0037) (Appendix A). In unadjusted Cox models, vaginal melanoma was associated with a higher risk of death (HR: 9.25 95% CI: 1.58–54, *p* = 0.014) in comparison to vulvar melanoma. However, after adjusting for age and disease stage, this effect was attenuated (HR: 5.70 95% CI: 0.68–48, *p* = 0.11), as most vaginal melanomas were also diagnosed at later stages.

In crude and adjusted analysis, both regional nodal involvement and/or distant metastases at diagnosis were significantly associated with worse survival (Figure 2 and Table 3). Survival worsened accordingly with increased disease stage with poorer prognosis in the nodal and metastatic stage (HR: 4.92 95% CI: 1.41–17.1 *p* = 0.01 and HR: 13.2 95% CI: 2.8–62 *p* = 0.001, respectively) in comparison to the localized stage (Figure 2 and Table 3).

Regarding treatment options and survival outcomes, surgically intervened patients displayed improved survival (Figure 2). In age and sex adjusted models, surgical intervention also appeared as protective (HR: 0.25 95% CI: 0.08–0.74, *p* = 0.012), though this protective effect was not maintained after adjusting for disease stage (HR: 0.44 95% CI: 0.12–1.57, *p* = 0.2) (Table 3).

## 4. Discussion

Our retrospective analysis of MM within a cohort of 35 patients provides valuable insights into the epidemiological aspects, clinical characteristics, and outcomes associated with this uncommon yet highly aggressive malignancy.

The observed higher incidence of MM among females compared to males (62.9% vs. 37.1%, *p* = 0.001) aligns with prior research [6,7,14,17]. This predominance in females can be attributed to the elevated occurrence of vulvovaginal melanoma, which represents the most frequently diagnosed subtype among women [4,5,8,18,19].

Our patient cohort exhibited a median age of 70 years at presentation, a statistic in keeping with existing literature [7,8,20]. This median age at the time of diagnosis highlights the predominant occurrence of MM among older individuals, emphasizing the need for proactive early detection initiatives, particularly in anatomical regions where MM tends to be more prevalent.

In our study, the most prevalent site was the vulvovaginal region, followed by the head and neck, and the digestive tract. Notably, this distribution differs from what is reported in the literature. Some studies suggest that the most common site is the head and neck, followed by vulvovaginal and anorectal locations [8,21,22], while others place anorectal ahead of vulvovaginal, with head and neck remaining the most frequent site [1,3,5,20,23,24]. Accurately estimating prevalence by location is challenging due to the rarity of this disease, leading to diverse prevalence percentages. Furthermore, the higher prevalence of vulvovaginalMM in our series may be attributed to our hospital’s Gynecological Oncology Department being a key referral center statewide, resulting in the referral of patients with this pathology to our center.

No statistically significant relationships were found between the MM location and the staging at diagnosis, in contrast to other studies that reported a higher incidence of lymph node involvement in cases located in the anorectal region [4,5,7]. Similarly, no association was observed between the location and survival, aligning with some studies [25], while others reported worse survival outcomes in anorectal locations [21,22,26], or the best survival outcomes in vulvovaginal location [21]. In our series, there is a notable poorer survival for vaginal MM compared to vulvar MM, consistent with findings in other studies in the literature [27,28]. However, statistical significance diminishes when adjusted for stage, demonstrating that patients with nodal or metastatic involvement at diagnosis exhibit poorer survival, regardless of vulvar or vaginal localization.

The diverse clinical presentations of MM across various anatomical sites highlight the diagnostic complexities associated with this condition. In our study, the most common symptoms included bleeding in vulvovaginal cases, rectorrhagia in anorectal cases, and varied symptoms depending on the specific location within the head and neck region. The high variability in clinical presentations of MM, its occurrence in anatomically challenging areas, and the potential for confusion with other pathologies, or the modesty of patients suffering from the condition delays medical consultations and, consequently, the diagnosis, often lead to disease progression, resulting in a more advanced stage at the time of diagnosis [1,4,5,20,22,29].

We did not find a correlation between smoking habit and the occurrence of MM. In the literature, this lack of association is evident in the case of vulvovaginal and anorectal MM [30]. However, some studies do identify an increased risk of head and neck MM among smokers [31,32], while others posit that there is no connection, at least between lip MM and tobacco use [33]. An interesting finding in our series regarding smoking is the association between smoking habits and the presence of metastases at the time of diagnosis, a phenomenon not previously reported in the MM literature. Nevertheless, it has been noted that active smoking is linked to decreased utilization of breast, colorectal, and cervical cancer screening services. Additionally, active smokers who do not undergo appropriate screening face significantly higher odds of being diagnosed with advanced-stage breast or colorectal cancer [34].

Our analysis revealed a significant correlation (*p* = 0.001) between younger age at diagnosis and a higher stage of MM. In particular, younger patients were inclined to exhibit more advanced disease stages. This age-staging relationship prompts intriguing inquiries concerning potential age-related distinctions in MM development, progression, or diagnostic methodologies that merit further investigation, as we did not find it documented in the existing literature.

We also observed that patients with distant metastases (*p* = 0.001) and regional nodal involvement (*p* = 0.012) exhibited significantly inferior survival outcomes compared to individuals with localized disease, in accordance with what has been reported in the literature [3,21,25,26,35]. These results underscore the critical importance in considering metastatic status and nodal involvement when assessing prognosis and tailoring treatment strategies for MM patients.

In our series, PET/CT stands out as the most frequently used imaging modality for detecting MM dissemination, consistent with its utility described in the majority of studies [3,30]. Selective sentinel lymph node biopsy (SLNB) was employed in only 15% of the patients. The role of SLNB in MM staging and management is currently under investigation, and its prognostic significance remains to be conclusively established [1,8].

Tumor thickness (i.e., Breslow depth) is a crucial prognostic factor in CM [17], but its significance in MM remains debated. Notably, our study reveals that a Breslow index of less than 5 mm is linked to better survival in MM. Patel et al. found that thicker tumors (exceeding 5 mm) in MM of the head and neck correlated with poorer survival in their study of 59 patients [36]. Conversely, no prognostic relationship was observed for the Breslow index in anorectal MM by Yeh et al. [37]. Tcheung et al. reported a significant connection between an increased Breslow index and worse survival in their study of 85 vulvovaginal MM patients [38]. While some studies suggest that a Breslow index greater than 5 mm is an independent prognostic factor [14], Altieri et al., in their analysis of 1824 patients with MM, did not identify the Breslow index as a prognostic factor, but they did note reduced survival in patients with an unknown Breslow index (comprising 70% of their cohort) [26]. Despite the small sample size in our series, this is an intriguing finding from our study to consider.

The *BRAF* mutations identified in our study align with findings from the existing literature [5,8,25,29,39,40,41]. Consistent with other series [29], our study did not establish a significant association between *BRAF* mutation status and either the localization or staging of MM. Moreover, it is important to note that the assessment of *c-KIT* mutations was limited to only six cases, thus precluding a comprehensive analysis of the relevance of *c-KIT* mutations in our cohort. Nevertheless, among the cases studied, 33% exhibited the mutation, consistent with the percentages reported in other series [8,20,22,25,41,42]. Additionally, one of the patients, who required systemic treatment, significantly benefited from targeted therapy using a tyrosine kinase inhibitor (imatinib) as a third-line treatment.

Further investigation with a larger sample size is needed to explore the potential clinical implications of these genetic alterations in MM.

Treatment strategies such as surgery, radiotherapy, and chemo-/immunotherapy for MM lack randomized trials for specific guidance [43]. Surgical removal of the tumor with clear margins is the primary approach, but achieving complete resection can be challenging due to tumor size, anatomical complexity, functional considerations, and proximity to vital structures [1,41]. Complete tumor excision has been identified as a prognostic factor [21]. In our series, surgically treated patients exhibited enhanced survival, although this statistical significance dissipates when adjusted for staging.

Radiotherapy is an option for adjuvant treatment or unresectable lesions [22]; however, its overall benefits remain uncertain. It may enhance local disease control without impacting overall survival [5,8,44].

MM differs molecularly from CM, showing lower rates of *BRAF* V600 alterations and tumor mutational burden but a higher rate of chromosomal aberrations [1,45,46]. Adjuvant systemic therapy options for MM are limited, with systemic therapy for CM often recommended [43]. MM’s lower mutational burden and reduced PD-L1 expression might explain their relatively poorer response to immunotherapy compared to CM [8,23,40].

Given the challenging outcomes and the lack of established treatments, participation in clinical trials is considered a standard of care for advanced-stage melanoma patients [43], including our center.

In our series, first-line systemic treatment involved immunotherapy with anti-PD-1 agents, consistent with previous research [1,22]. Combining anti-PD-1/anti-CTLA-4 agents is also a potential option, particularly for selected patients, although this can lead to increased toxicity [23,30,40,42]. For cases with *c-KIT* or *BRAF* mutations, targeted therapies may be considered [7,22,30,42], although durable responses remain unproven [23].

Within our series, we have identified a consistent set of clinical and pathological characteristics in MM that independently predict overall survival (OS) through multifactorial analysis. These include tumor thickness, as previously mentioned, and the presence of metastases in regional lymph nodes or distant sites. Our observations align with findings from other studies [3,22,25,26,30], underscoring the paramount importance in early detection and timely intervention in MM cases, particularly those involving regional nodal involvement or distant metastases at the time of diagnosis.

Regarding OS, we report a 2-year survival rate of less than 50% and a 5-year survival rate below 25%. These figures are in line with rates reported in the existing literature [1,3,4,26,41,45]. The fact that a significant proportion of patients in our cohort succumbed to this malignancy underscores the aggressive nature of MM.

The findings of our series are in concordance with those of most previous studies, rendering them applicable and transferable to actual clinical practice in other specialized tertiary hospitals. Nonetheless, it is crucial to acknowledge the inherent limitations in our study. These limitations include its retrospective and single-center design, which limits its external validity, makes it vulnerable to bias and incomplete data recollection, and the relatively modest sample size, which yielded imprecise estimates at times. Due to the small sample size, the high heterogeneity of cases, and the substantial variability in treatment regimens, it was not feasible to perform a comparative analysis of survival outcomes based on treatment combinations nor systemic therapies in this study.

## 5. Conclusions

In conclusion, our retrospective analysis of MM has provided valuable insights into this rare yet highly aggressive malignancy. Significantly poorer survival outcomes were observed in cases with distant metastasis and regional nodal involvement compared to localized disease. Our findings suggest that tumor thickness, specifically a Breslow index of less than 5 mm, could serve as a potential prognostic factor in MM, correlating with improved survival rates, and underscores the correlation between more advanced stages at diagnosis and younger age. Finally, we highlight the presence of a notably high percentage of mutations in *c-KIT*.

The treatment landscape for MM remains devoid of standardized guidelines, with surgery being the primary approach despite inherent challenges tied to tumor size and anatomical complexities. Immunotherapy emerges as a prevalent systemic treatment, though questions persist regarding optimal combination therapies and associated toxicities. The consideration of targeted therapies becomes crucial in cases involving mutations in *BRAF* or *c-KIT*. To advance our understanding of prognostic factors and treatment modalities in MM, future research endeavors involving larger cohorts are essential. Such efforts hold promise for enhancing our knowledge and management of MM.

## Figures and Tables

**Figure 1 cancers-16-00227-f001:**
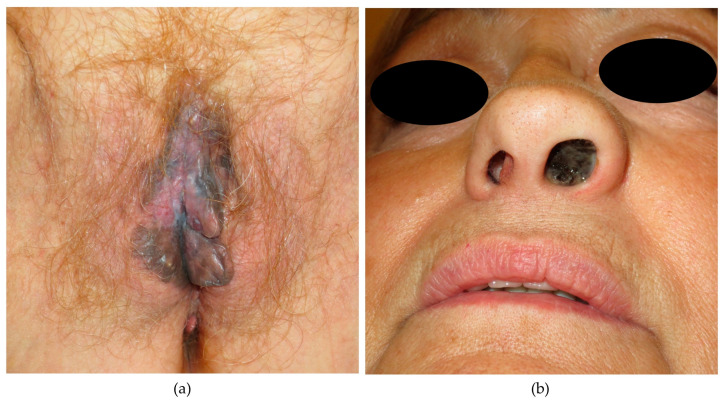
Clinical images of two patients affected with MM. (**a**) A 68-year-old woman presented with a heterochromatic lesion measuring 7 × 5 cm, involving the labia majora, minora and the clitoris, without palpable lymphadenopathies. A biopsy revealed melanoma with extensive superficial ulceration; (**b**) A 69-year-old woman presented with a left nasal wing bulging associated with a blackish mass visible through the left nostril. Biopsy confirmed the presence of melanoma, and PET-CT scans revealed paratracheal and supraclavicular lymphadenopathies.

**Figure 2 cancers-16-00227-f002:**
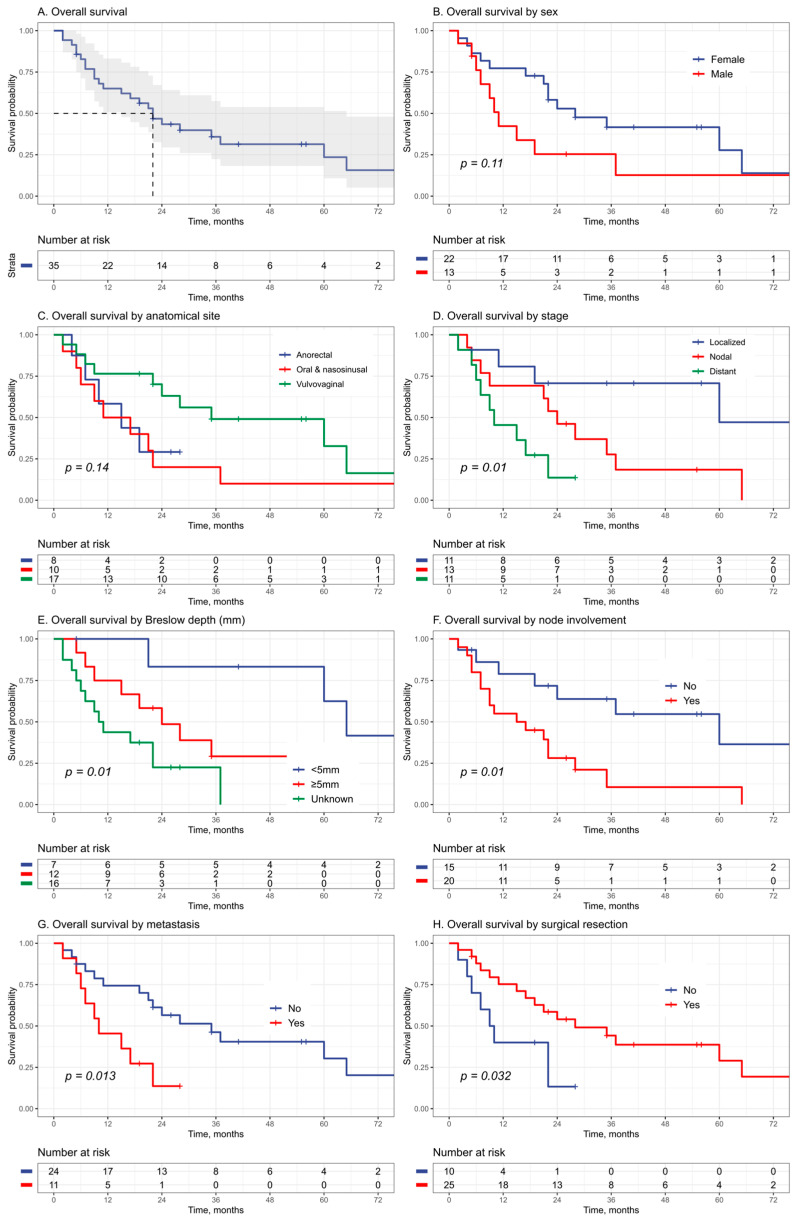
Kaplan–Meier curves for overall survival. Curves are shown for the overall population and gray area indicates 95% confidence interval (subset (**A**)) and for different clinical and tumor characteristics (subsets (**B**–**H**)).

**Table 1 cancers-16-00227-t001:** Patient and tumor characteristics by anatomical site.

	Anorectal * (N = 8)	Head and Neck (N = 10)	Vulvovaginal (N = 17)	Total (N = 35)	*p*-Value
Sex					<0.001
Female	2 (25.0%)	3 (30.0%)	17 (100.0%)	22 (62.9%)	
Male	6 (75.0%)	7 (70.0%)	0 (0.0%)	13 (37.1%)	
Age at diagnosis, years					0.912
Median (Q1, Q3)	67.5 (59.8, 82.2)	72.0 (64.5, 80.2)	68.0 (58.0, 82.0)	70.0 (60.5, 82.0)	
Smoking, ever					0.173
No	1 (14.3%)	5 (55.6%)	9 (56.2%)	15 (46.9%)	
Yes	6 (85.7%)	4 (44.4%)	7 (43.8%)	17 (53.1%)	
Unknown	1	1	1	3	
Immunosuppression					0.126
No	8 (100.0%)	8 (80.0%)	17 (100.0%)	33 (94.3%)	
Yes	0 (0.0%)	2 (20.0%)	0 (0.0%)	2 (5.7%)	
Disease Stage					0.567
Localized	2 (25.0%)	3 (30.0%)	6 (35.3%)	11 (31.4%)	
Nodal	2 (25.0%)	4 (40.0%)	7 (41.2%)	13 (37.1%)	
Distant	4 (50.0%)	3 (30.0%)	4 (23.5%)	11 (31.4%)	
Breslow depth, mm					0.324
Median (Q1, Q3)	10.5 (5.2, 15.0)	3.0 (1.5, 6.0)	5.2 (2.5, 8.7)	5.5 (2.9, 9.0)	
Unknown	4	7	5	16	
Breslow depth, mm					0.503
Unknown	4	7	5	16	
<5 mm	1 (25.0%)	2 (66.7%)	4 (33.3%)	7 (36.8%)	
≥5 mm	3 (75.0%)	1 (33.3%)	8 (66.7%)	12 (63.2%)	
*BRAF* mutations					0.784
No	8 (100.0%)	8 (88.9%)	14 (87.5%)	30 (90.9%)	
Yes	0 (0.0%)	1 (11.1%)	2 (12.5%)	3 (9.1%)	
Unknown	0	1	1	2	
Nodal involvement					0.625
No	2 (25.0%)	5 (50.0%)	8 (47.1%)	15 (42.9%)	
Yes	6 (75.0%)	5 (50.0%)	9 (52.9%)	20 (57.1%)	
Nodal status assessment technique					0.628
CT	1 (16.7%)	0 (0.0%)	3 (33.3%)	4 (20.0%)	
MRI	2 (33.3%)	1 (20.0%)	1 (11.1%)	4 (20.0%)	
PET/CT	3 (50.0%)	2 (40.0%)	4 (44.4%)	9 (45.0%)	
SLNB	0 (0.0%)	2 (40.0%)	1 (11.1%)	3 (15.0%)	
Unknown	2	5	8	15	
Metastasis					0.453
No	4 (50.0%)	7 (70.0%)	13 (76.5%)	24 (68.6%)	
Yes	4 (50.0%)	3 (30.0%)	4 (23.5%)	11 (31.4%)	
Surgical resection					0.796
No	3 (37.5%)	2 (20.0%)	5 (29.4%)	10 (28.6%)	
Yes	5 (62.5%)	8 (80.0%)	12 (70.6%)	25 (71.4%)	
Radiotherapy					0.867
No	7 (87.5%)	7 (70.0%)	13 (76.5%)	27 (77.1%)	
Yes	1 (12.5%)	3 (30.0%)	4 (23.5%)	8 (22.9%)	
Systemic therapy					0.710
No	2 (25.0%)	3 (30.0%)	7 (41.2%)	12 (34.3%)	
Yes	6 (75.0%)	7 (70.0%)	10 (58.8%)	23 (65.7%)	
Number of systemic treatment lines					0.935
Median (Q1, Q3)	1.0 (0.8, 1.2)	1.0 (0.2, 2.0)	1.0 (0.0, 3.0)	1.0 (0.0, 2.0)	
Follow-up/survival time, months					0.095
Median (Q1, Q3)	12.5 (6.5, 20.8)	14.0 (6.8, 21.8)	28.0 (19.0, 55.0)	21.0 (8.0, 35.0)	
Death					0.247
No	3 (37.5%)	1 (10.0%)	7 (41.2%)	11 (31.4%)	
Yes	5 (62.5%)	9 (90.0%)	10 (58.8%)	24 (68.6%)	

Q1: first quartile, Q3: third quartile. * The anorectal category includes one case of esophageal MM.

**Table 2 cancers-16-00227-t002:** Management of MM by anatomical site.

Anatomical Site	Treatments Received				
	Surgery	RT	Systemic Treatment		
			First Line	Second Line	Third Line
Vulvovaginal	12/17 (70.6%)	4/17 (23.5%)	10/17 (58.9%) Anti-PD-1: 3 (30%) Chemotherapy **: 2 (20%) Anti-PD-1 + LAG-3 inh: 2 (20%) BRAF inh + MEK inh: 1 (10%) Interferon: 1 (10%) Anti-PD-1 + IDO inh: 1 (10%)	7/17 (41.1%) Anti-CTLA-4: 3 (42.8%) Anti-PD-1: 2 (28.6%) Chemotherapy **: 1 (14.3%) Anti-PD-1 + Anti-CTLA-4: 1 (14.3%)	5/17 (29.4%) Anti-CTLA-4: 2 (40%) Chemotherapy **: 1 (20%) Pan-RAF inh: 1 (20%) Tyrosine kinase inh (imatinib): 1 (20%)
Anorectal ***	5/8 (62.5%)	1/8 (12.5%)	6/8 (75%) Anti-PD-1: 2 (25%) Clinical trial *: 2 (25%) Chemotherapy **: 1 (12.5%) Anti-PD-1 + anti-CTLA-4: 1 (12.5%)	2/8 (25%) Anti-CTLA-4: 2 (100%)	0/8 (0%)
Head and neck	8/10 (80%)	3/10 (30%)	7/10 (70%) Anti-PD-1: 3 (42.8%) Chemotherapy **: 1 (14.3%) BRAF inh + MEK inh: 1 (14.3%) Anti-PD-1 + LAG-3 inh: 1 (14.3%) Anti-PD-1 + anti-CTLA-4: 1 (14.3%)	4/10 (40%) Chemotherapy **: 1 (25%) Anti-PD-1: 1 (25%) Anti-CTLA-4: 1 (25%) Clinical trial *: 1 (25%)	1/10 (10%) Clinical trial *: 1 (100%)
Total	25/35 (71.4%)	8/35 (22.9%)	23/35 (65.7%) Anti-PD-1: 8 (34.8%) Chemotherapy **: 4 (17.4%) Anti-PD-1 + LAG-3 inh: 3 (13%) BRAF inh + MEK inh: 2 (8.7%) Clinical trial *: 2 (8.7%) Anti-PD-1 + anti-CTLA-4: 2 (8.7%) Interferon: 1 (4.3%) Anti-PD-1 + IDO inh: 1 (4.3%)	13/35 (37.1%) Anti-CTLA-4: 6 (46.2%) Anti-PD-1: 3 (23.1%) Chemotherapy **: 2 (15.4%) Anti-PD-1 + anti-CTLA-4: 1 (7.7%) Clinical trial *: 1 (7.7%)	6/35 (17.1%) Anti-CTLA-4: 2 (33.3%) Chemotherapy **: 1 (16.7%) Pan-RAF inh: 1 (16.7%) Tyrosine kinase inh (imatinib): 1 (16.7%) Clinical trial: 1 (16.7%)

* Clinical trial with blinded treatment. ** Chemotherapy included treatments based on dacarbazine, platinum and/or taxanes. Abbreviations: RT: radiotherapy; inh: inhibitor; PD-1: programmed death protein 1; LAG-3: lymphocyte activation gene 3; CTLA-4: cytotoxic T-lymphocyte antigen 4; IDO: indoleamine 2,3-dioxygenase. *** The anorectal category includes one case of esophageal MM.

**Table 3 cancers-16-00227-t003:** Association of clinical and tumor characteristics with all-cause mortality in Cox regression models.

	Crude Model			Model 1 *			Model 2 **		
Characteristic	HR ^1^	95% CI ^1^	*p*-Value	HR ^1^	95% CI ^1^	*p*-Value	HR^1^	95% CI ^1^	*p*-Value
Age at diagnosis, 10 y	0.93	(0.69, 1.27)	0.7	0.95	(0.70, 1.29)	0.7	1.41	(0.93, 2.12)	0.093
Sex			0.12			0.13			0.083
Female	1.00	—		1.00	—		1.00	—	
Male	1.94	(0.85, 4.43)	0.12	1.92	(0.84, 4.39)	0.12	2.16	(0.92, 5.05)	0.076
Anatomical site			0.15			0.5			0.7
Anorectal ***	1.00	—		1.00	—		1.00	—	
Head and Neck	1.09	(0.36, 3.34)	0.9	1.10	(0.35, 3.48)	0.9	1.53	(0.44, 5.29)	0.5
Vulvovaginal	0.47	(0.15, 1.43)	0.2	0.50	(0.11, 2.39)	0.4	1.07	(0.22, 5.31)	>0.9
Surgical resection			0.044			0.013			0.2
No	1.00	—		1.00	—		1.00	—	
Yes	0.37	(0.15, 0.93)	0.035	0.25	(0.08, 0.74)	0.012	0.44	(0.12, 1.57)	0.2
Breslow depth, mm	1.35	(1.12, 1.63)	<0.001	1.47	(1.14, 1.89)	<0.001	1.51	(1.13, 2.01)	<0.001
Nodal Involvement			0.008			<0.001			
No	1.00	—		1.00	—				
Yes	3.23	(1.29, 8.07)	0.012	7.02	(2.30, 21.5)	<0.001			
Metastasis			0.021			0.018			
No	1.00	—		1.00	—				
Yes	3.04	(1.22, 7.59)	0.017	3.61	(1.24, 10.5)	0.018			
Disease stage			0.009			0.002			
Localized	1.00	—		1.00	—				
Nodal	3.13	(0.98, 10.0)	0.055	4.92	(1.41, 17.1)	0.012			
Distant	6.47	(1.79, 23.5)	0.004	13.2	(2.82, 62.0)	0.001			

^1^ HR = hazard ratio, CI = confidence interval. * Model 1: adjusted for age and sex. ** Model 2: Model 1 further adjusted by localized, nodal or distant disease. *** The anorectal category includes one case of esophageal MM.

## Data Availability

The data that support the findings of this study are available from the corresponding author upon reasonable request.

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
