# Peer review of "Mucosal Melanoma Clinical Management and Prognostic Implications: A Retrospective Cohort Study"

_cancers, 2024, doi:10.3390/cancers16010227_

Round 1
Reviewer 1 Report
Comments and Suggestions for Authors
1. This is not a cohort study. It is a retrospective convenience sample.
2. n is small, so that all the negative findings could be the result of type II error, Study would greatly benefit from larger sample size obtained by recruiting more medical cancers.
3. similarly, Tabel 2 is fairly meaningless because it reflects small sample size and institutional and time bias
4. "PCR" needs to be spelled out the first time it is used
5. The most interesting thing about this paper are the findings on PCR; that BRAF mutations were found in only 9% of mucosal melanomas, while 1/3 had c-kit mutations. That is actually the most solid and novel finding given the small n used for the demographic and descriptive part of the paper. And yet, these findings are not in the abstract. Consider shortening the paper to just this observation.
Comments on the Quality of English Language
Overall good, some minor grammar issues
Author Response
Thank you for your valuable comments. Please see below the answer one by one.
- This is not a cohort study. It is a retrospective convenience sample.
Thank you for carefully reviewing our methodology. For this study we selected all mucosal melanoma cases diagnosed or treated at our hospital that had a pathological confirmation from our Pathology department. This type of study is frequent in Oncology and Epidemiology and is in fact a type of cohort study, more precisely a registry-based cohort study (eg. 10.1016/j.breast.2017.12.015). We are glad to specify it as a registry-based cohort study if you think it may help clarify the nature of the study cohort (this has been added at the simple summary, abstract and methodology).
We understand how you may view it as a convenience sample as we are selecting cases that had histological confirmation and therefore were readily available to us (convenience). However, it is unlikely that a case may have been omitted from this cohort as pathological confirmation for melanoma diagnosis is carried out routinely in our center.
- n is small, so that all the negative findings could be the result of type II error, study would greatly benefit from larger sample size obtained by recruiting more medical cancers.
We are well aware of our limited sample size and share the reviewer’s concern about the potential impact of a Type II error. As it is widely known, mucosal melanoma is an uncommon malignancy in caucasian populations and it was therefore hard to obtain a large sample size. While we understand the benefits of a multicenter approach in expanding sample size, we will explore collaborative opportunities with other medical centers for future research initiatives.
Regarding the influence of a Type II error, we acknowledge the inherent limitations of our sample size and its impact on statistical power. However, attributing all negative findings to Type II error oversimplifies the multifaceted nature of research outcomes. As a mainly descriptive study, formal statistical power calculations were not a priority. We did, however, provide confidence intervals to the readers to ensure that the uncertainty of our estimates is adequately informed.
Perhaps the size of our series is not as extensive as in other studies; however, we intentionally confined our series to the past 10 years. This period corresponds to the emergence of systemic treatments for melanoma, marking a profound shift in its management over the last decade, significantly impacting patient prognosis. To minimize era bias and ensure uniformity in our cohort and subsequent survival conclusions, we delimited our study to the past 10 years. Notably, existing literature lacks cohort studies encompassing diverse mucosal melanoma locations. While studies focusing on singular locations and spanning over 20 years exist (e.g., 10.1046/j.1525-1438.2001.01043.x; 10.1016/j.joms.2015.05.021; 10.1002/hed.10019; 10.1016/j.jaad.2011.11.921; 10.1016/j.joms.2016.03.008), none consolidates various mucosal melanoma sites into a single cohort.
- similarly, Table 2 is fairly meaningless because it reflects small sample size and institutional and time bias.
Our center serves as a national referral center for managing patients with metastatic melanoma, drawing patients from across the peninsular region. We conduct open Phase I, II, and III clinical trials, enrolling patients who stand to benefit from these treatments. Notably, as current clinical guidelines lack directly approved treatments for patients with mucosal melanoma, our trials offer an avenue for potential therapeutic options in this specific context. In our opinion, Table 2 offers an overview on how difficult the management of these patients is.
- "PCR" needs to be spelled out the first time it is used
Thank you for the appreciation. We have modified it.
- The most interesting thing about this paper are the findings on PCR; that BRAF mutations were found in only 9% of mucosal melanomas, while 1/3 had c-kit mutations. That is actually the most solid and novel finding given the small n used for the demographic and descriptive part of the paper. And yet, these findings are not in the abstract. Consider shortening the paper to just this observation.
We find this series to be highly intriguing as there are very few works that comprehensively gather different types of mucosal melanomas on a global scale. However, considering truncating the work solely to this conclusion seems somewhat inappropriate as the mutation was only studied in 6 cases, resulting in a small sample size. Nevertheless, we strongly agree that this finding is compelling, and therefore, we have incorporated it into the abstract, discussion, and conclusion sections. Additionally, among these cases where the mutation was positive (2 out of 6 cases, accounting for 33%), one of them notably benefited from targeted therapy with imatinib as a third-line treatment, which is an interesting point worthy of consideration (see Table 2).

Reviewer 2 Report
Comments and Suggestions for Authors
Given the limited amount of studies on mucosal melanoma, this is a pretty valuable and interesting retrospective cohort study.
One comment is in section 4. discussion, line 289 to 294, the author said "... the most prevalent site was vulvovaginal region, ........ Some studies suggst that the most common site is the head and neck,... while others place anorectal ahead of vulvovaginal,...." The author should discuss what causes this discrepancy.
Author Response
Thank you for your valuable comments. Please see below the answer.
One comment is in section 4. discussion, line 289 to 294, the author said "... the most prevalent site was vulvovaginal region, ........ Some studies suggest that the most common site is the head and neck,... while others place anorectal ahead of vulvovaginal,...." The author should discuss what causes this discrepancy.
Thank you very much for your comment. We will proceed to include the following information in the article, as we agree it is relevant. Accurately estimating prevalence by location is challenging due to the rarity of this disease, leading to diverse prevalence percentages. Furthermore, the higher prevalence of vulvovaginal mucosal melanomas in our series may be attributed to our hospital's Gynecological Oncology Department being a key referral center statewide, resulting in the referral of patients with this pathology to our center.

Round 2
Reviewer 1 Report
Comments and Suggestions for Authors
None